# Genistin: A Novel Potent Anti-Adipogenic and Anti-Lipogenic Agent

**DOI:** 10.3390/molecules25092042

**Published:** 2020-04-27

**Authors:** Yae Rim Choi, Jaewon Shim, Min Jung Kim

**Affiliations:** 1Research Division of Food Functionality, Korea Food Research Institute, Wanju 55365, Koreajwshim@kfri.re.kr (J.S.); 2Department of Food Science and Engineering, Ewha Womans University, Seoul 03760, Korea

**Keywords:** genistin, soy isoflavones, anti-adipogenesis, anti-lipogenesis

## Abstract

Soy isoflavones are popular ingredients with anti-adipogenic and anti-lipogenic properties. The anti-adipogenic and anti-lipogenic properties of genistein are well-known, but those of genistin and glycitein remain unknown, and those of daidzein are characterized by contrasting data. Therefore, the purpose of our study was to investigate the effects of daidzein, glycitein, genistein, and genistin on adipogenesis and lipogenesis in 3T3-L1 cells. Proliferation of 3T3-L1 preadipocytes was unaffected by genistin and glycitein, but it was affected by 50 and 100 µM genistein and 100 µM daidzein for 48 h. Among the four isoflavones, only 50 and 100 µM genistin and genistein markedly suppressed lipid accumulation during adipogenesis in 3T3-L1 cells through a similar signaling pathway in a dose-dependent manner. Genistin and genistein suppress adipocyte-specific proteins and genes, such as peroxisome proliferator-activated receptor γ (PPARγ), CCAAT-enhancer-binding protein α (C/EBPα), and adipocyte binding protein 2 (aP2)/fatty acid-binding protein 4 (FABP4), and lipogenic enzymes such as ATP citrate lyase (ACL), acetyl-CoA carboxylase 1 (ACC1), and fatty acid synthase (FAS). Both isoflavones also activate AMP-activated protein kinase α (AMPKα), an essential factor in adipocyte differentiation, and inhibited sterol regulatory element-binding transcription factor 1c (SREBP-1c). These results indicate that genistin is a potent anti-adipogenic and anti-lipogenic agent.

## 1. Introduction

Obesity is the status of increased adipose tissue mass and one of the metabolic diseases leading to public health problems worldwide [1]. Obese people have two internal characteristics: an increase in the fat cell number (hyperplasia) and the adipose cell size (hypertrophy) [2]. Adipogenesis is the process of cell differentiation from preadipocytes into mature adipocytes leading to hyperplasia [3]. Lipogenesis is the process of fatty acid and triglyceride synthesis by the conversion of acetyl-CoA into triglycerides for storage as fat leading to hypertrophy [4]. According to in vitro studies using 3T3-L1 cells, hormones or drugs such as insulin and β-adrenoceptor agonists stimulate fibroblast-like 3T3-L1 cells causing differentiation into adipocyte-like cells and lipid accumulation [5,6,7,8,9]. Incubating 3T3-L1 preadipocytes with the MDI media (a mixture of 3-isobutyl-1-methylxanthine (M), dexamethasone (D), and insulin (I)) promotes a synchronized cell cycle, mitotic clonal expansion (MCE), and adipogenesis [10,11].

The molecular mechanisms associated with adipogenesis and lipogenesis have been extensively investigated. Adipogenic-specific transcription factors such as CCAAT-enhancer-binding protein α (C/EBPα), peroxisome proliferator-activated receptor γ (PPARγ), and adipocyte binding protein 2 (aP2)/fatty acid-binding protein 4 (FABP4) play a crucial role in adipocyte formation and lipid production [12,13]. Lipogenic enzymes, including ATP citrate lyase (ACL), acetyl-CoA carboxylase 1 (ACC1), and fatty acid synthase (FAS), are associated with lipogenesis, inducing fatty acid synthesis, and lipid accumulation [14]. AMP-activated protein kinase (AMPK) and sterol regulatory element-binding transcription factor 1c (SREBP-1c) are some other proteins involved in adipogenesis and lipogenesis. Although AMPK is a key factor for regulating cellular energy balance, it is also involved in adipogenesis by regulating SREBP-1c, which can modulate adipogenic-specific factors and lipogenic enzymes [14,15].

Soybean contains numerous isoflavones that prevent obesity and related metabolic disorders, including inflammation, certain types of cancers, cardiovascular disease, and hyperglycemia [16,17,18,19,20,21]. Soy isoflavones usually exist as glycosides. Glycoside in isoflavones is released from isoflavones by fermentation, following which soy isoflavone aglycones are formed. Aglycones of genistin, daidzin, and glycitin are genistein, daidzein, and glycitein, respectively. Daidzein, genistein, glycitein, and genistin are the major isoflavones in soybean (*Glycine max*) and soy products. According to the Phenol Explorer databases, genistin is found in soybeans (4.55 mg/100 g) and soy products, such as soy flours (103.28 mg/100 g), soy pastes (89.42 mg/100 g), and roasted soybeans (67.76 mg/100 g), which showed higher levels than daidzein (soybeans, 0.58 mg/100 g; soy flours, 2.58 mg/100 g; soy pastes, 12.11 mg/100g; roasted soybeans, 5.17 mg/100 g), glycitein (soybeans, 0.47 mg/100 g; soy flours, 1.90 mg/100 g; soy pastes, 6.56 mg/100 g; roasted soybeans, 6.45 mg/100 g), and genistein (soybeans, 0.54 mg/100 g; soy flours, 3.62 mg/100 g; soy pastes, 3.50 mg/100 g; roasted soybeans, 4.89 mg/100 g) [22]. These chemicals have structural similarities (Scheme 1) [17,23,24,25,26]. With the backbone consisting of 7-hydroxyisoflavones, daidzein has an additional hydroxy group at position 4′, and glycitein is substituted by a methoxy group at position 6′ in daidzein. Genistein has an additional hydroxy group at positions 5′ and 4′, and genistin is substituted by 7-*O*-β-D-glucoside from genistein. Although the chemical structures of isoflavones are similar, their biological activities and activity levels and the mechanisms underlying the activities of each isoflavone are not always identical. Genistein, but not daidzein, has been shown to protect plant cells from oxidative degradation by chelating copper (II) ions [27]. Both daidzein and genistein affect angiogenesis, oxidative stress, cell proliferation, and cell death [28,29,30,31,32]. However, genistein exerts more profound effects on adipogenesis, metastasis, lipolysis, and epigenetic modifications than daidzein. Both daidzein and genistein are estrogen receptor agonists, but genistein is stronger than daidzein [33]. Both daidzein and genistein have been demonstrated to inhibit prostate cancer proliferation and induce cell cycle arrest through modulation of cyclin-dependent kinases (CDKs) via different mechanisms; daidzein and genistein arrest the G_0_/G_1_ phase and G_2_/M phase, respectively [34,35]. Accordingly, soy isoflavones may have different adipogenic properties. In previous studies, the anti-adipogenic efficacy of genistein has been well-demonstrated, whereas few reports have been published on the adipogenic effects of genistin and glycitein in 3T3-L1 preadipocytes till date and the studies investigating the adipogenic effect of daidzein have reported contradictory results [29,31,32,36,37,38,39]. In addition, the mechanism underlying the role of genistin and glycitein in the differentiation of 3T3-L1 cells has not been studied yet.

In the present study, we investigated the effects of four soy isoflavones, daidzein, glycitein, genistein, and genistin, on the proliferation of 3T3-L1 preadipocytes and lipid accumulation in 3T3-L1 adipocytes. The mechanisms involved in adipogenesis and lipogenesis during the differentiation of 3T3-L1 preadipocytes were also evaluated by monitoring adipogenic-specific factors, lipogenic factors, and upstream regulators (AMPKα and SREBP-1c) using Western blotting and real-time polymerase chain reaction (RT-PCR).

## 2. Results

### 2.1. Cytotoxicity of Soy Isoflavones in 3T3-L1 Preadipocytes

The toxicity of soy isoflavones was investigated before monitoring the effects on adipocyte differentiation. 3T3-L1 preadipocytes were incubated with 0, 25, 50, or 100 µM daidzein, glycitein, genistein, or genistin for 24 and 48 h, and the cytotoxicity was analyzed using CCK-8 and live/dead cell assays (Figure 1). After 24 h, 25–100 μM glycitein, genistein, and genistin and 25–50 μM daidzein did not affect the proliferation of 3T3-L1 preadipocytes. However, 100 μM daidzein significantly reduced the viability of 3T3-L1 cells (p < 0.05). Comparison of live/dead cell imaging showed that there was no difference between the control and 100 μM daidzein-treated cells. Therefore, 100 μM daidzein was not cytotoxic, but reduced cell proliferation. After 48 h, 25–100 μM glycitein and genistin did not affect cell proliferation. Daidzein (25–50 μM) had no significant effect on cell viability, but 100 μM daidzein suppressed cell viability. Genistein (50–100 μM, but not 25 μM) also reduced cell viability by 12.5 and 28.8% dose-dependently. However, the live/dead levels in 100 μM daidzein- or 25–50 μM genistein-treated cells were similar to those in the control. Therefore, daidzein and genistein were not cytotoxic, but reduced cell proliferation. Thus, up to 100 μM of the four isoflavones were non-cytotoxic to 3T3-L1 cells.

### 2.2. Effect of Soy Isoflavones on 3T3-L1 Adipocyte Differentiation

Inhibition of lipid accumulation by isoflavones was quantitatively and qualitatively measured after adipocyte differentiation (Figure 2). Genistein and genistin (50 and 100 μM) significantly inhibited lipid accumulation in 3T3-L1 adipocytes concentration-dependently. The reduction rates of lipid accumulation in 50 and 100 μM genistin-induced 3T3-L1 adipocytes were 21.7 and 69.2%, respectively, and those in 50 and 100 μM genistein-induced adipocytes were 37.2 and 81.9%, respectively. Images after Oil Red O staining also supported the quantitative result that genistein and genistin reduced lipid droplet formation as compared to the control. Daidzein and glycitein did not affect intracellular lipid accumulation. Thus, only genistin and genistein inhibited lipid accumulation in 3T3-L1 adipocytes.

### 2.3. Inhibitory Effects of Genistin and Genistein on the Protein and Gene Expression of Adipogenic-Specific Factors During Differentiation of 3T3-L1 cells

Among the four compounds, only genistin and genistein, which suppressed lipid accumulation, were used to demonstrate the changes in adipogenic-specific proteins (Figure 3A). During the differentiation from preadipocytes to adipocytes, C/EBPα, PPARγ, and aP2/FABP4 protein expression levels were increased. The protein expression levels of C/EBPα, PPARγ, and FABP4 in 25–100 μM genistin- or genistein-treated 3T3-L1 adipocytes were decreased dose-dependently. These results were identical in messenger RNA (mRNA) levels, as shown in Figure 3B-D. Gene expression levels of C/EBPα, PPARγ, and aP2 in 25–100 μM genistin- and genistein-treated 3T3-L1 adipocytes were also attenuated dose-dependently. Protein and gene expression levels of adipogenic-specific factors in both genistin- and genistein-treated cells were similar. Consequently, genistin and genistein could inhibit adipogenesis through similar mechanisms.

### 2.4. Suppressive Effects of Genistin and Genistein on the Protein and Gene Expression of Lipogenic Enzymes

Based on the effects of genistin and genistein on lipid accumulation, the changes in gene expression of lipogenic enzymes in genistin- or genistein-treated 3T3-L1 cells were analyzed (Figure 4). The mRNA expression levels of ACL, ACC1, and FAS were significantly suppressed by 25, 50, and 100 μM genistin and genistein; however, ACL expression level was unaffected by 25 μM genistein treatment. The gene expression of the three factors in genistin- or genistein-treated 3T3-L1 cells decreased dose-dependently. The expression of lipogenic enzyme genes in both genistin- and genistein-treated cells was similar. Thus, the mechanisms of lipogenesis in genistin- and genistein-treated cells may be similar.

### 2.5. Modulation of AMPK Activity and SREBP-1c Expression by Genistin and Genistein

AMPKα is related to not only energy homeostasis, but also lipid metabolism, affecting lipogenic genes including SREBP-1c. Therefore, the Western blot analysis was performed to examine AMPKα and phospho-AMPKα (p-AMPKα) expression levels, and RT-PCR was used to estimate SREBP-1c mRNA levels (Figure 5). The protein expression levels of AMPKα were similar in all samples, but those of p-AMPKα in 50 and 100 μM genistin- and genistein-treated 3T3-L1 cells decreased. SREBP-1c gene expression levels decreased dose-dependently. SREBP-1c mRNA expression was significantly attenuated by 100 μM genistein and 50 and 100 μM genistein. AMPKα and SREBP-1c protein levels in genistin-treated cells were similar to those in genistein-treated cells. Therefore, it is likely that the efficacies and mechanisms of genistin and genistein against adipogenesis and lipogenesis are similar.

## 3. Discussion

This study explored the efficacy of four isoflavones, daidzein, glycitein, genistein, and genistin, and identified the mechanism of genistin against adipogenesis and lipogenesis for the first time. In addition, we showed similar mechanisms between genistin and genistein during differentiation (Figure 6). The proliferation of 3T3-L1 preadipocytes was not affected by 25–100 μM glycitein and genistin treatment for 48 h; however, it was decreased by 100 μM daidzein treatment for 24–48 h and 50 and 100 μM genistein treatment for 48 h. As a result of lipid accumulation, anti-adipogenic properties were observed only in genistin- and genistein-treated cells. Genistin and genistein downregulated the expression of adipogenic-specific factors and lipogenic enzyme genes dose-dependently. In addition, both isoflavones activated AMPK, a cellular energy homeostasis regulator, and inhibited SREBP-1c dose-dependently. Particularly, the protein and gene expression levels of all monitored factors in genistin- and genistein-treated cells were similar.

Various factors, including AMPK, are related to adipogenesis and lipogenesis. AMPK is the main component for regulating cellular energy balance and cell cycle; however, it has been revealed to regulate adipogenesis and lipogenesis [16,40,41,42,43,44,45]. Activated AMPK directly or indirectly modulates downstream factors, such as proliferation pathway-related proteins, metabolic enzymes, and transcription/translation factors, which further suppresses the synthesis of triglycerides, cholesterol, and fatty acids in lipid metabolism, causing fatty acid uptake and β-oxidation [46]. Among AMPK subtypes, AMPKα is mainly expressed in adipose tissues and regulates fat energy homeostasis. Activated AMPKα directly triggers the phosphorylation of precursor SREBP-1c (Ser372) and inhibits the proteolysis of precursor SREBP-1c into a mature form [47,48]. SREBP-1c, one of the three SREBP isoforms, is a pro-adipogenic transcription factor and a valuable target for metabolic diseases, because it is mainly related to fatty acid synthesis and lipogenesis [49]. The regulation of the AMPK and/or SREBP-1c signaling pathway by soy isoflavones was studied previously. Genistein inhibits adipogenesis in 3T3-L1 cells by stimulating intracellular reactive oxygen species (ROS) release and subsequently activating AMPK [36,38]. It has also been reported to activate AMPK/ACC in 3T3-L1 cells [38]. Daidzein and genistein trigger an increase in phospho-AMPK (Thr172) and decrease in SREBP-1c expression in rats [16]. This AMPK/SREBP1c signaling has been proved by applying Compound C, an AMPK inhibitor, with daidzein or genistein. When diet-induced obese rats were treated with Compound C and either daidzein or genistein, the protein level of phospho-AMPK was higher and that of SREBP-1c was lower than that in Compound C-treated rats. Our results also showed that genistein upregulated AMPKα phosphorylation and downregulated SREBP1 mRNA expression in 3T3-L1 adipocytes. The effects of genistin on AMPKα phosphorylation and SREBP1 mRNA expression in 3T3-L1 adipocytes have not been studied before, but the activities of genistin on both factors were similar to that of genistein, which induced activation of AMPKα phosphorylation and inhibition of SREBP1 mRNA expression. Therefore, the activity of genistin is expected to be similar to genistein.

Suppressed mature SREBP-1c reduces the expression of adipogenic-specific factors and lipogenic transcription factors [50,51,52]. C/EBPα belongs to the C/EBP family, which includes C/EBPβ and C/EBPδ. C/EBPs are abundantly expressed at different stages of differentiation of 3T3-L1 cells [53]. The expression levels of C/EBPβ and C/EBPδ are transiently increased at the early stage of differentiation, which promotes adipogenesis, whereas those of C/EBPα and PPARγ are the highest at the later stages of differentiation. PPARγ, which is predominant in adipogenic differentiation among the PPAR family, is expressed by SREBP-1c induction [54]. PPARγ and C/EBPα cause mutual expression and cooperate in the activation of some adipogenic genes, such as aP2/FABP4. SREBP-1c also downregulates ACL, ACC1, and FAS, which are fatty acid synthesis-related genes. ACL converts citrate to acetyl-CoA; subsequently, ACC1 converts acetyl-CoA to malonyl-CoA. Finally, fatty acids are produced by FAS from malonyl-CoA. Many studies have demonstrated that SREBP-1c directly binds to the ACL promoter, ACC1 promoter, and regulatory factor (E-box motif) in the FAS gene enhancer [51,52,55]. Therefore, downregulated mature SREBP-1c causes a decrease in ACL, ACC1, and FAS.

Genistein also inhibits adipogenesis and lipogenesis via these mechanisms. Park et al. (2009) showed that genistein reduces lipid accumulation in a concentration-dependent manner by decreasing SREBP-1c, adipogenic-specific transcription factors (PPARγ, C/EBPα, aP2, and glycerol-3-phosphate dehydrogenase), FAS, and lipolysis factors (hormone-sensitive lipase, HSL) in primary human preadipocytes [39]. In 3T3-L1 adipocytes, genistein also downregulates diverse factors including PPARγ, C/EBPα, ACC, FAS, FABP4, HSL, chemerin, resistin, and GLUT4 [36,56,57]. In diet-induced obese male rats, soy isoflavones, including genistein, ameliorate visceral fatty acid metabolism by downregulating fat synthesis (ACC1, ACC2, ACL, and FAS) and upregulating fat hydrolysis (ATGL and HSL) [16]. Furthermore, human adipose tissue-derived mesenchymal stem cells are disturbed to differentiation by genistein via the Wingless and Int/β-catenin (Wnt/β-catenin) signaling pathway [58]. Consistent with other data, the present study showed almost identical effects of genistein on adipogenic factors and lipogenic enzymes during differentiation of 3T3-L1 cells. Genistein suppressed adipogenic factors, including C/EBPα, PPARγ, and aP2/FABP4, and, furthermore, lipogenic transcription factors, including ACL, ACC, and FAS, dose-dependently.

The suppressive effect of genistin on lipid accumulation in 3T3-L1 cells was identified for the first time. Our results demonstrated that 50 and 100 μM genistin decreases lipid accumulation, identical to the same concentrations of genistein. In previous studies, only low concentration of genistin (20 μM) was investigated in 3T3-L1 cells, which showed no inhibitory effect on lipid accumulation, and 10^−8^–10^−5^ M genistin attenuates the differentiation of bone marrow stromal cells from osteoblast adipocytes [59,60]. Moreover, the anti-adipogenic and anti-lipogenic mechanisms of genistin in 3T3-L1 cells were almost identical to those of genistein. Genistin activated AMPK and suppressed SREBP-1c, adipogenic factors, including C/EBPα, PPARγ, and aP2/FABP4, and lipogenic enzymes, including ACL, ACC, and FAS, dose-dependently, all of which have similar responses to genistein at the same dose range. That is, upstream regulators, including AMPKα and SREBP-1c, were modulated by genistin at 100 μM and by genistein at 50 and 100 μM. The suppressive effects of genistin and genistein on the gene expression of C/EBPα, PPARγ, and aP2/FABP4 were identical in that both isoflavones at 50 and 100 μM inhibited C/EBPα expression and at 25~100 μM inhibited PPARγ and aP2/FABP4 expression. ACL expression was also suppressed by genistin (25~100 μM) and genistein (50 and 100 μM) and the gene expression of ACC and FAS was also inhibited by genistin and genistein at 25, 50, and 100 μM. The effective concentrations and mechanisms of genistein have been reported in previous studies [36,38,56,57]. Based on the anti-adipogenic and anti-lipogenic mechanism of genistein, the mechanisms of adipogenesis and lipogenesis can be summarized as shown in Figure 6 and genistin appeared to inhibit adipogenesis and lipogenesis by suppressing phosphorylation of AMPK in 3T3-L1 cells. Therefore, genistin could also serve as a potent anti-adipogenic and anti-lipogenic agent, similar to genistein.

Daidzein has shown contradictory effects on adipogenesis in many studies. He et al. (2016) reported that 50–200 μM daidzein reduced the proliferation of 3T3-L1 preadipocytes due to mitotic clonal expansion and attenuated fat deposition and adipogenesis via downregulation of two nuclear transcription factors, C/EBPα and PPARγ, and of other differentiation-associated genes (LPL, ACC, FADS2, FAS, and SCD1) [29]. In contrast, Seo et al. (2013) and Yang et al. (2019) reported that 20-80 μM daidzein does not inhibit MDI-induced adipogenesis of 3T3-L1 adipocytes [60,61]. Several other studies have shown that daidzein inhibits adipogenesis and obesity by regulating obesity-related genes and energy-associated hormones [19,29,37,61,62]. Of the two opposing effects, our results supported that where daidzein has no effect on adipogenesis. A 100 μM daidzein solution attenuated the proliferation of 3T3-L1 preadipocytes, but did not affect 3T3-L1 differentiation.

In summary, this study provided new evidence that genistin modulates AMPK/SREBP-1c signaling and inhibits adipogenesis of 3T3-L1 adipocytes by reducing C/EBPα, PPARγ, and aP2/FABP4, and lipogenesis by modulating ACL, ACC, and FAS. Adipogenesis inhibition by genistin needs to be established in vivo, but it is expected to have anti-obesity effects, because the key mechanisms of genistin for anti-adipogenesis were the same as those for genistein. Therefore, we suggest that genistin can provide a potential therapeutic approach for preventing obesity and obesity-related metabolic disorders.

## 4. Materials and Methods

### 4.1. Reagents

Dimethyl sulfoxide (DMSO), dexamethasone (DEX), 3-isobutyl-1-methylxanthine (IBMX), insulin, Oil Red O, and four soy isoflavones (daidzein, genistein, genistin, and glycitein) were purchased from Sigma-Aldrich (St. Louis, MO, USA). Structures of daidzein, genistein, genistin, and glycitein are shown in Figure 1. The Dulbecco’s modified Eagle’s medium (DMEM), fetal bovine serum (FBS), bovine calf serum (BCS), and phosphate buffered saline (PBS) were purchased from Welgene Inc. (Daegu, Korea). The live/dead cell imaging kit (488/570) was purchased from Invitrogen (Molecular Probes, Life Technologies Corp., CA, USA). Cell counting kit-8 (CCK-8) was purchased from Enzo Life Sciences Inc. (Farmington, NY, USA). Antibodies to peroxisome proliferator-activated receptor-γ (PPARγ), CCAAT-enhancer-binding protein α (C/EBPα), fatty acid-binding protein (FABP4), adenosine monophosphate-activated protein kinase α (AMPKα), and phospho-AMPKα were purchased from Cell Signaling Technology (Bedford, MA, USA). β-Actin was purchased from Santa Cruz Biotechnology (Santa Cruz, CA, USA).

### 4.2. Cell Culture, Differentiation, and Reagents

3T3-L1 preadipocytes were purchased from the American Type Culture Collection (ATCC, USA). Cells were cultured in the DMEM supplemented with 10% bovine calf serum and 1% penicillin/streptomycin (P/S) at 37 °C with 5% CO_2_. To induce differentiation, culture media of confluent preadipocytes (day 0) were changed to the MDI differentiation media (the DMEM containing 10% FBS, 1% P/S, 0.5 mM IBMX, 1 µM dexamethasone, and 1 µg/mL insulin) and differentiated for 36 h. At this point, cells were treated with various concentrations of four soy isoflavones in the MDI differentiation media. After 36 h, the culture media were replaced with maintenance media (the DMEM supplemented with 10% FBS, 1% P/S, and 1 µg/mL insulin) with four soy isoflavones and changed every two days. Differentiation was analyzed based on the expression of adipogenic markers and appearance of lipid accumulation. Differentiation was completed on day 7 or 9.

### 4.3. Cell Cytotoxicity Analysis

3T3-L1 preadipocytes were seeded into 96-well plates. After 24 h, preadipocytes were treated with daidzein, genistein, genistin, or glycitein at concentrations of 0, 25, 50, and 100 μM. After 24 h or 48 h of incubation, the media were replaced and the CCK-8 solution was added to each well. After incubation for 2 h, absorbance was measured at 450 nm, with the reference absorbance at 650 nm. Cell viability (%) was calculated as follows:(1)$$\mathrm{Cell}\;\mathrm{viability}\;\left(\%\right)=\frac{\left(\mathrm{Absorbance}\;\mathrm{of}\;\mathrm{the}\;\mathrm{sample}-\mathrm{Reference}\;\mathrm{absorbance}\right)}{\mathrm{Absorbance}\;\mathrm{of}\;\mathrm{the}\;\mathrm{control}}\times 100$$

### 4.4. Live/Dead Imaging

3T3-L1 preadipocytes were seeded into 96-well microplates and treated with daidzein and genistein at 5% CO_2_, 37 °C for 24 h and 48 h. The preadipocytes were stained with the live/dead cell imaging kit as described by the manufacturer’s protocol. In brief, an equal volume of 2 × working solution was added to the cells and incubated for 15 min at 25 °C. After the treatment, the cells were visualized using fluorescent microscopy. For live cell image (green), the excitation and emission wavelengths were 488 and 515 nm, respectively. For dead cell image (red), the excitation and emission wavelengths were 570 and 602 nm, respectively.

### 4.5. Oil Red O (ORO) Staining

3T3-L1 preadipocytes were seeded into 6-well culture plates and cultured for 2 days after reaching confluence. Then, media were changed to the MDI differentiation media and cells were treated with four soy isoflavones. After 36 h, the differentiation media were changed to maintenance media containing each soy isoflavone, which were replaced every 2 days. At day 7, 3T3-L1 cells were fixed with 4 % formaldehyde for 1 h and washed two times with the PBS. The cells were rinsed with 60% isopropanol for 5 min and then dried completely. The cells were stained in the Oil Red O working solution (0.3 g in 100 mL isopropanol) for 10 min and then washed with H_2_O. The cells were examined in H_2_O under a microscope. Oil red O quantification was performed by extracting the dye by 100% isopropanol and optical density (OD) values were measured in 96-well plates at a wavelength of 500 nm. The percentage of ORO-stained cells was calculated as (A_isoflavones_ – A_blank_)/(A_vehicle_ – A_blank_) × 100%

### 4.6. Western Blot Analysis

After 7 days, cells were lysed in the radioimmunoprecipitation assay (RIPA) buffer with protease and phosphatase inhibitors for 45 min and then centrifuged at 13,500 rpm for 15 min at 4 °C. Protein concentrations were determined using the bicinchoninic acid (BCA) protein assay. Equal amounts of proteins were separated using sodium dodecyl sulfate (SDS)–polyacrylamide gel electrophoresis (PAGE). Proteins were transferred to polyvinylidene difluoride (PVDF) membranes (Bio-Rad), blocked with 5% bovine serum albumin (BSA) in the Tris-Buffered Saline with Tween 20 (TBST) with 0.05% Tween 20 for 1 h, and incubated overnight with primary antibodies against PPARγ, C/EBPα, FABP4, AMPKα, and p-AMPKα at 1:1000 dilution at 4 °C. After washing in TBST, the membranes were incubated with secondary antibodies at 1:2000 dilution for 2 h. Proteins were detected using an enhanced chemiluminescence (ECL) Western blotting detection kit and exposed to X-ray.

### 4.7. Real-Time Polymerase Chain Reaction (RT-PCR)

mRNAs of C/EBPα, PPARγ, aP2, SREBP-1c, ACC1, ACL, and FAS were estimated by the RT-PCR method. Total RNA was isolated from 3T3-L1 cells using the TRIzol reagent (Invitrogen) according to the manufacturer’s protocol. Total RNA (2 μg) was converted to complementary DNA (cDNA) using a SuperScript III Real-Time (RT) kit (Invitrogen) and RT-PCR was performed using the QuantStudio 6 Flex Real-Time Polymerase Chain Reaction (PCR) system (GE Healthcare) with the primers described in Table 1.

### 4.8. Statistical Analysis

One-way analysis of variance (ANOVA) was used to determine statistical significance of the differences between values of various experimental and control groups with the Tukey’s post-hoc test. Data are expressed as the mean ± standard error (n ≥ 3). *p*-values < 0.05 were considered statistically significant. All analyses were performed using the GraphPad Prism 5 software (GraphPad Software, San Diego, CA, USA).

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
