# Peer review of "Genistin: A Novel Potent Anti-Adipogenic and Anti-Lipogenic Agent"

_molecules, 2020, doi:10.3390/molecules25092042_

Round 1
Reviewer 1 Report
In the paper entitled “Genistin: A novel potent anti-adipogenic and antilipogenic agent” authors investigate the potential anti lipo/adipogenic properties of flavonoid Genistin revealing its potential beneficial role.
Paper is well written, rationale is clear, figures legends are exhaustive and data flow is convincing.
However in discussion section, Authors should focalize Genistin biochemical properties section to better justify its Genistein-like behaviour as potential anti adipogenic features; moreover authors should provide more information on Genistin content in various food or diet in order to give more transnationality to their good paper.
Reviewer 2 Report
The study by Yae Rim Choi et al entitled “Genistin: A novel potent anti-adipogenic and anti-lipogenic agent” has been reviewed. The study, evaluates the effects of four soy isoflavones, daidzein, glycitein, genistein, and genistin, on the proliferation of 3T3-L1 preadipocytes and lipid accumulation in 3T3-L1 adipocytes.
The manuscript is clear and well written, however, the preliminary toxicity analysis of soy isoflavones seems to be not in line with the experimental model since they performed it for a time of exposure of 24 and 48 hours while, the experimental protocol to evaluate the mechanisms involved in adipogenesis and lipogenesis during the differentiation of 3T3-L1 preadipocytes by monitoring adipogenic-specific factors, lipogenic factors, and upstream regulators requires an exposure time of at least 7 days.
The authors reported:
“The toxicity of soy isoflavones was investigated before monitoring the effects on adipocyte differentiation. 3T3-L1 preadipocytes were incubated with 0, 25, 50, or 100 μM daidzein, glycitein, genistein, or genistin for 24 and 48 h, and the cytotoxicity was analysed using CCK-8 and live/dead cell assays.”(…………………………………….) “Differentiation was analysed based on the expression of adipogenic markers and appearance of lipid accumulation. Differentiation completed on day 7 or 9.”
Please clarify this point and provide clear data on this aspect.
Moreover, it is appropriate to evaluate the toxicity of these compounds also on a human cell line since the cell line under examination has mouse origin.
Round 2
Reviewer 2 Report
Although my requests were not accepted in total, the replies were sufficiently reasoned by the authors. In any case, it would be appropriate that in your next investigative models you expected to adopt the same experimental conditions for all tests, where this is possible, especially for the preliminary analysis of toxicity since a compound can exert its cytotoxic effects with a chronic exposure rather than with acute exposure. However, the paper is suitable for publication.